# Constitutively Active Androgen Receptor in Hepatocellular Carcinoma

**DOI:** 10.3390/ijms232213768

**Published:** 2022-11-09

**Authors:** Emma J. Montgomery, Enming Xing, Moray J. Campbell, Pui-Kai Li, James S. Blachly, Allan Tsung, Christopher C. Coss

**Affiliations:** 1Division of Pharmaceutics and Pharmacology, College of Pharmacy, The Ohio State University, 496 W. 12th Ave., Columbus, OH 43210, USA; 2Division of Medicinal Chemistry and Pharmacognosy, College of Pharmacy, The Ohio State University, 496 W. 12th Ave., Columbus, OH 43210, USA; 3Division of Hematology, Department of Internal Medicine, The Ohio State University, 410 W. 12th Ave., Columbus, OH 43210, USA; 4Division of Surgical Oncology, Department of Surgery, University of Virginia Hospital, 1300 Jefferson Park Ave., Charlottesville, VA 22903, USA

**Keywords:** liver cancer, nuclear hormone receptor, androgen receptor splice variants, androgen receptor C-terminal truncated isoforms, antiandrogens, androgen receptor degraders

## Abstract

Hepatocellular carcinoma (HCC) is the predominant type of liver cancer and a leading cause of cancer-related death globally. It is also a sexually dimorphic disease with a male predominance both in HCC and in its precursors, non-alcoholic fatty liver disease (NAFLD)/non-alcoholic steatohepatitis (NASH). The role of the androgen receptor (AR) in HCC has been well documented; however, AR-targeted therapies have failed to demonstrate efficacy in HCC. Building upon understandings of AR in prostate cancer (PCa), this review examines the role of AR in HCC, non-androgen-mediated mechanisms of induced AR expression, the existence of AR splice variants (AR-SV) in HCC and concludes by surveying current AR-targeted therapeutic approaches in PCa that show potential for efficacy in HCC in light of AR-SV expression.

## 1. Background

### 1.1. Sexual Dimorphism in Hepatocellular Carcinoma

Hepatocellular carcinoma (HCC) is the predominant form of liver cancer making up 90% of cases and is currently the fourth most lethal form of cancer as well as the sixth most common cancer worldwide [1]. HCC exhibits sexual dimorphism with men having an increased susceptibility of between two and seven-fold higher than women regardless of disease etiology [2]. This sexual dimorphism is also a feature of several HCC precursors including non-alcoholic fatty liver disease (NAFLD) and non-alcoholic steatohepatitis (NASH) [3]. Up to date data on HCC and related liver cancers in the United States show liver and intrahepatic bile duct cancer presently have an overall 5-year survival rate of 20.8%, breaking down to 20.6% for men and 21.5% for women. Delay and age adjusted incidence rates of liver and intrahepatic bile duct cancer in the United States for 2019 are 9.7 in 100,000 overall with men having a higher incidence of 14.6 per 100,000 compared to 5.5 per 100,000 for women. The most recent U.S. mortality rates for liver and intrahepatic bile duct cancer are 6.5 in 100,000 overall with men again having the higher rate at 9.4 in 100,000 while women have a mortality rate of 4.1 per 100,000. It is projected that there will be 41,260 new cases in 2022 and 30,520 deaths from liver and intrahepatic bile duct cancer in the United States in 2022 [4]. Globally, higher HCC incidence and mortality are found with the highest rates being in East Asia and Africa, and disease etiology varying widely by region [1].

### 1.2. Androgen Receptor Expression and Role in Disease Progression

Based upon the discrepancy in NASH/NAFLD and HCC incidence between the sexes, the role of the male nuclear sex hormone receptor, the androgen receptor (NR3C4/AR), in HCC progression has been investigated. The AR is a transcription factor that operates via ligand-activation by binding to androgens (e.g., testosterone (T), 5α-dihydrotestosterone (DHT)) resulting in nuclear translocation, dimerization, and binding to androgen response elements (AREs) across the genome allowing regulation of target gene transcription. The AR consists of 3 functional domains; the N-terminal domain (NTD), which contains the binding domains for the transcriptional machinery and cofactors needed for transcriptional regulation, the DNA-binding domain (DBD), which binds to AREs within the genome to regulate target gene transcription, and the ligand-binding domain (LBD) which binds to circulating androgens leading to AR activation and nuclear translocation (Figure 1) [5]. Early studies on AR in HCC found that mice without hepatic AR expression had slower carcinogen-mediated HCC progression and smaller tumors compared to mice with AR even in the presence of similar levels of testosterone, indicating that hepatic AR action and not circulating androgens themselves are the key feature in dimorphic HCC progression [6]. However, the differences in HCC incidence between the sexes are not readily explained by AR expression alone as shown by the Sex-Associated Gene Database, where males have slightly higher average levels of hepatic AR expression but differences in hepatic AR expression between males and females are not statistically significant [7]. Additionally, the AR plays a role in Hepatitis B virus (HBV) induced HCC. The HBV genome contains a functional ARE within its promoter such that the AR is able to directly upregulate transcription of HBV, thus promoting HBV-related hepatocarcinogenesis [8]. This feed forward mechanism in HBV-mediated HCC is consistent with even larger differences in male and female HCC rates in regions where chronic HBV infection is endemic [1,9,10].

Though data are mixed, the balance of early studies on AR levels in HCC tumors found significantly higher AR protein expression when compared with adjacent normal liver tissues and that increased AR protein expression was correlated with increased tumor recurrence and reduced overall survival [6,11]. However, a more recent study found that while cytoplasmic AR protein was not significantly different between tumor and adjacent liver tissue, nuclear AR protein levels were significantly elevated in the tumor as compared to adjacent normal liver tissue [12]. Early studies into AR mRNA expression within HCC found high between subject variability of AR mRNA and higher AR mRNA levels in tumor as compared to adjacent normal tissue. However, later studies failed to observe a difference in AR mRNA expression between tumor and peritumoral tissue [11]. Recently, Acosta-Lopez et al. found that while both higher AR mRNA and protein expression were correlated with higher overall survival, higher AR activity as measured by androgen responsive genes showing differential expression between HCC histological grades was associated with a worse prognosis [13]. In support of the more recent reports, our survey of publicly available AR data show that, relative to other cancers, AR protein (Figure 2A) and mRNA (Figure 2B) expression in HCC is relatively high. Additionally, AR mRNA levels decrease in liver cancer tissue relative to adjacent normal controls (Figure 2B). An examination of AR’s relationship to overall survival from either Reverse-Phase Protein Array (RPPA) or RNA-Seq of liver cancer patients and associated survival data from The Cancer Genome Atlas (TCGA) shows a similar result with higher AR protein (Figure 2C) and mRNA (Figure 2D) expression being significantly correlated with higher overall survival [14]. Collectively, these data show that AR mRNA expression alone is not sufficient to explain sexual dimorphism in HCC outcomes as higher mRNA and protein expression are correlated with improved survival. Instead, these findings suggest that activated and nuclear localized AR as well as measures of AR activity are better correlated with poor HCC outcomes as opposed to general AR mRNA or protein expression. We further explore this complex relationship between AR expression and activity and clinical outcomes in subsequent sections.

### 1.3. AR and Epithelial–Mesenchymal Transition

The AR has been implicated in Epithelial–Mesenchymal Transition (EMT) regulation in the early literature on cancer. One study looking at the role of AR and EMT in breast, prostate, and other cancer cells found that AR directly downregulated E-cadherin expression through an ARE leading to EMT and metastases [18]. Additional findings within prostate cancer showed that overexpression of AR leads to cells undergoing EMT, and overexpression of AR splice variant 7 (AR-V7), a constitutively active AR variant, (Figure 1) not only led to EMT but also to stem cell gene signatures further supporting a role for AR in EMT and metastases [19]. Within the context of HCC, transcriptomic analyses of 24 HCC cell lines from the Cancer Cell Line Encyclopedia (CCLE) database revealed that higher AR expressing cell lines were enriched for expression of EMT-related genes. Additionally, an ARE was found in the promoter for SNAI2, the gene encoding the transcriptional factor SLUG, which is implicated in regulation of EMT genetic programs. AR regulation of SNAI2 was found to be liver cancer specific with no correlation between AR and SNAI2 levels in normal tissue expression data [20]. One study examining the role of AR in metastatic activity in HCC found that higher AR expressing cell lines had increased lamellipodia and RAC1 expression. They also found that AR-mediated upregulation of Rac1 expression increased the level of intra- and extra-hepatic (pulmonary) metastases in HCC [21].

### 1.4. Failed Therapeutic Approaches Targeting the AR

Based upon the frank male bias in HCC incidence and mortality along with abundant pre-clinical support for the role of AR in HCC, AR inhibition as a therapy for HCC has been evaluated in a series of clinical trials using combined AR-axis inhibition approaches routinely deployed in prostate cancer (PCa). Grimaldi et al. conducted a double-blind trial with a two-by-two design to test the effectiveness of nilutamide, a first generation anti-androgen [22], and a luteinizing hormone-releasing hormone (LHRH) agonist, a peptidomimetic molecule that suppresses gonadal androgen biosynthesis, both alone and in tandem against a placebo. The trial concluded that while there were few side effects associated with hormonal therapy, the therapies did not show any significant clinical benefit to patients with unresectable HCC [23]. A later trial involving males with late-stage HCC compared a group of patients treated with flutamide, another first-generation anti-androgen, leuprolide (also known as leuprorelin), an LHRH agonist, and tamoxifen against patients treated only with tamoxifen. They similarly found no benefit to survival in the group treated with flutamide and leuprolide and instead found a nearly significant decrease in survival in the multi-treatment group as compared to the group treated only with tamoxifen [24]. Finally, a recent clinical trial evaluated enzalutamide, a second generation anti-androgen with improved AR antagonist properties [25], alone or in combination with sorafenib in advanced HCC patients. While patients were able to tolerate the combination of enzalutamide and sorafenib, a pharmacokinetic drug–drug interaction reduced the effectiveness of sorafenib when given in combination with enzalutamide and enzalutamide offered no improvement either in combination with sorafenib or alone over sorafenib monotherapy [26]. Despite the improvements in AR antagonists developed for PCa throughout the course of these trials from nilutamide to flutamide to enzalutamide, none were able to elicit a response in HCC. It is critical to highlight that within the context of PCa, many approaches beyond steroid-competitive AR antagonism and suppression of androgen biosynthesis are currently being investigated as alternative approaches to targeting the AR-axis [27]. The remainder of this review will focus on new insights into the complexities of AR signaling in HCC and how these new AR-targeted approaches might be effectively deployed in HCC.

### 1.5. Reconciling the Role of AR in HCC with the Failure of Anti-Androgens

The contribution of AR signaling to HCC progression has been supported with abundant clinical and pre-clinical evidence showing its correlation with faster disease progression, tumor burden, disease prognosis, overall survival, and as a mediator of metastasis. However, the failure of antiandrogen therapy to improve patient outcomes seems to conflict with an understanding of ligand-dependent AR signaling. To reconcile the clinical failure of androgen-targeted therapy and the overwhelming evidence supporting a role for AR in HCC, there are several explanations to consider. Expression of the AR may be dissociated from circulating androgen levels and could result from feedback from other signaling pathways in HCC. The difference between expression and activity is also a key distinction as increased expression of AR may not necessarily mean an increase in AR activity or vice versa. Finally, examining mechanisms of ligand-independent constitutive activation of AR could decouple AR signaling from androgen dependent AR activation. The examination of these alternative explanations has resulted in a better understanding of the role of AR signaling in HCC and potential therapeutic avenues.

## 2. Alternative Mechanisms of AR Overexpression

### 2.1. mTOR Overexpression and Signaling in HCC

Androgen-independent induction of AR expression is one possible resistance mechanism to anti-androgen therapy. AR expression is in part mediated by androgens which can both downregulate AR expression to suppress further AR signaling in the presence of higher levels of androgen or as shown in several HCC cell lines, androgens can upregulate AR expression [28,29]. However, there are several mechanisms of androgen-independent AR overexpression including the mammalian target of rapamycin (mTOR) and its associated signaling which plays a role in the induction of AR expression through multiple pathways. However, mTOR plays a key role in HCC in its own right. In around half of all HCCs, the PI3K/AKT/mTOR signaling pathway was found to be hyperactivated and in 15% of HCCs phosphorylated mTOR expression is increased. Additionally, PTEN, which can activate PI3K/AKT/mTOR signaling, is mutated in about half of HCC cases. Activation of the mTOR signaling pathway is correlated with metabolic activity, increased cell proliferation, and support of tumor survival [30]. Despite the evidence of mTOR’s involvement in HCC, trials of the potent mTOR inhibitor Everolimus failed to show clinical benefit [31]. However, the failure of mTOR inhibition as a monotherapy in HCC may also be related to the interplay of mTOR and AR signaling.

### 2.2. AKT-mTOR and AR Crosstalk

mTOR engages in a complex signaling pathway that can induce AR expression. Zhang et al. found that mTORC1 plays a role in both inhibiting AR degradation as well as upregulating the translocation of AR to the nucleus (Figure 3A). Once nuclear, genomic AR signaling then leads to increased expression of FKBP5, a factor that in combination with PHLPP1 inhibits AKT expression and by extension mTORC1 formation (Figure 3B). Inhibiting AR increases FKBP5 and induces feedback activation of AKT-mTOR signaling. Interestingly, their understanding of mTOR and AR crosstalk conflicts with findings in PCa as they found that mTOR positively regulates the transcriptional activity of AR; while in PCa, mTOR is thought to negatively regulate AR. These data highlight the need to carefully evaluate AR signaling in each respective cancer type. Additionally, they found that higher AR nuclear localization, and therefore increased active AR, was associated with more advanced HCC and lower overall survival [12].

In a recent follow-up study from the same group, the mechanism behind mTORC1’s ability to inhibit AR degradation and promote AR nuclear localization was further elucidated. Ren et al. reported that mTORC1 phosphorylates AR at S96, in the amino-terminal domain, which enhances AR stability, translocation to the nucleus, and increased activity of AR independently of ligand (Figure 3C). Increased phosphorylation at this site is an independent predictor of reduced overall survival for HCC patients. Additionally, they found that elevated AKT expression promoted tumor growth at a faster rate in male mice than females [32].

### 2.3. Lipogenesis Driven Constitutive AR Activity

Lipogenesis also demonstrates a key linkage to AR in HCC progression as demonstrated by Cheng et al. who observed that AR activity and transcription of AR could be affected by dysregulated lipogenesis resulting from high fat diets that induce HCC. Specifically, diacylglycerols (DAGs) were shown to activate AKT which then in turn would activate AR (Figure 3D) demonstrating a further linkage between the AKT/mTOR pathway and AR in non-alcoholic fatty liver disease-mediated HCC. Additionally, they found that AR activation could be reversed through inhibition of fatty acid synthase (FASN) [33]. In PCa, a similar linkage has been demonstrated. Zadra et al. developed an FASN inhibitor and found that it was able to suppress both the expression and activity of full-length AR and AR-V7, a truncated ligand-independent AR splice variant well-characterized in PCa [34]. These studies support a role for lipogenesis in the induction of AR expression and activity and provide evidence that this pathway operates similarly in both PCa and HCC.

### 2.4. CCRK-AR-mTOR Pathway

In addition to mTOR’s signaling relationship with AR, cell cycle-related kinase (CCRK) has also been shown to play a role in mTOR signaling through AR. In a study conducted by Feng et al., CCRK was established as a direct AR transcriptional target through unbiased ChIP-Chip studies in HCC cells and confirmed to have high AR binding in its regulatory region. They proposed a signaling pathway whereby AR upregulates CCRK which in turn upregulates ß-catenin to exert effects on the cell cycle and on overall cellular proliferation (Figure 3E). This proposed signaling pathway also contains a feedback loop in which ß-catenin upregulates AR. To help confirm this pathway, they examined patient tumors to show that AR, CCRK, and ß-catenin were concurrently overexpressed in HCC tumors and that this overexpression was correlated with more advanced tumor stage and lower overall survival [35].

Another alternative mechanism for the induction of AR expression was developed by Sun et al. within the context of obesity-related HCC. They advanced an AR, STAT3, and CCRK feedback loop in which CCRK helps to incite STAT3 and AR interactions and their combined localization to androgen response elements (ARE) within the CCRK promoter upregulates CCRK expression. In addition, CCRK indirectly activates mTOR signaling pathways which in turn promotes HCC progression (Figure 3E). This was supported by their finding that all the elements within their proposed pathway are upregulated in clinical HCC samples when compared with non-cancerous liver tissue [36].

### 2.5. FAK-Mediated Signaling

Apart from CCRK related-signaling, another proposed pathway offers a further link between ß-catenin and AR signaling via cooperation with focal adhesion kinase (FAK). The authors, Shang et al., elucidated a direct link between overexpression of FAK paired with ß-catenin mutations and increased levels of AR activation within HCC. They posit that FAK acts by enhancing ß-catenin binding to the AR promoter thereby upregulating AR expression. Notably, they found that carcinogenesis occurring through this pathway is likely AR dependent but androgen independent [37]. This corroborates other prior findings including the failure of anti-androgen clinical trials and could help to explain why attempts to regulate AR signaling by competitively antagonizing androgen binding with anti-androgens or suppressing gonadal androgen synthesis in HCC may have failed to produce clinical benefit.

These mechanisms of inducing AR expression while not providing a comprehensive overview of the AR interactome and signaling pathways in HCC, provide valuable insight into how AR interacts with other oncogenic signaling pathways and their potential to upregulate the AR-axis. Databases such as BioGRID have compiled a wide array of AR interactors which provides additional support for the need to better understand AR and how it fits into the broader array of dysfunctional signaling pathways within HCC [38].

## 3. Difference between AR Expression and Activity

Several key studies have examined the role of AR in HCC by seeking to differentiate AR expression from activity and androgen-dependent AR signaling from androgen-independent AR signaling. Analyses of the HCC cohort in TCGA have helped illuminate the role of AR activity and expression in hepatocarcinogenesis. Acosta-Lopez et al. showed that while total AR mRNA levels are positively correlated with increased overall survival in this data set, gene regulation resulting from known AR transcriptional activity are negatively correlated with overall survival. This insinuates that AR activation rather than AR expression is key to HCC progression. As a caveat, the gene sets used in these analyses to determine AR activity were derived from PCa and in the future, more HCC specific measures of AR expression may be more valuable [13].

## 4. Alternative Splicing as a Means of Constitutive Activity

### 4.1. Decoupling of AR Activity from Androgen Binding

In addition to evidence of induced AR expression, data show that AR activity is not always a direct result of androgen binding in HCC (Figure 4A). At least in mice, the presence of AR protein and not circulating androgen levels is responsible for sexual dimorphism in HCC [6]. While activation of AR by mTOR phosphorylation is one possible explanation, androgen-independent, constitutively active AR could also explain the failure of anti-androgen therapy in HCC patients [32].

### 4.2. Variant Splicing in HCC

One mechanism for androgen-independent AR signaling involves constitutively active AR splice variants capable of driving AR transcriptional programs in the absence of androgen binding. Lee et al. report that mRNA splicing factors can be expressed differentially within HCC leading to alternative splicing patterns that can then contribute to multiple oncogenic pathways. The hijacking of various splicing factors through differential expression, silencing, or changes in the RNA binding proteins present can replace normative splicing patterns with alternative patterns resulting in proteins with new, similar or even opposite function when compared to the canonical splice form [39]. Given the high levels of variant splicing reported in HCC [40], it follows that alternative splicing of the AR could be a source of androgen-independent AR signaling in HCC.

### 4.3. Variant AR Splicing Is a Function of AR Overexpression

Most of the current knowledge surrounding the existence of AR splice variants (AR-SVs) and their activity is framed within the context of PCa, in which the androgen receptor and its splice variants play an established role. Within PCa, anti-androgens have been shown to rapidly upregulate splice variants of AR, such as AR variant 7 (AR-V7), which contain cryptic exons and premature stop codons preventing them from expressing the ligand-binding domain and ultimately rendering them androgen independent [41]. Many AR variants retain the ability to drive AR-mediated transcription and remain active even when in an androgen depleted environment (Figure 4B). Additionally, AR-V7 can also promote and facilitate nuclear localization of full length AR (AR-FL) even in the absence of endogenous androgens (Figure 4C) [42]. While AR and AR-V7 have an overlapping set of genes that they can regulate, AR-SVs do have unique transcriptional activity which may be responsible for the regulation of some oncogenic processes within PCa [43]. Variant AR can also modulate transcriptional activities even at lower expression levels than AR-FL [44].

This understanding of AR-SVs and their role in PCa led to an investigation of a similar role for AR-SVs in HCC. We confirmed the existence of AR-SVs within HCC and determined through analyses of TGCA data that 78% of HCC patients had intra-tumoral AR-SV expression and subsequently discovered that in the patients with the most abundant AR mRNA, AR-SVs accounted for roughly one quarter of expressed AR. Using direct comparisons to well characterized PCa models, these results showed that AR-SVs’ abundance within HCC were comparable to AR-SVs in PCa and AR-SV expression in HCC was higher than in normal liver controls. Additionally, we established both AR-FL and AR-V7 action can promote cell migration and invasion through EMT signaling in agreement with multiple reports showing AR signaling plays a role in cancer metastasis. We found that the SNAI2 gene, whose product is an established mediator of EMT (e.g., SLUG protein), is likely a direct target of the AR [20]. Finally, we expanded on current findings surrounding AR and mTOR signaling by examining AR-SV specific interactions with mTOR. Our data agreed with Zhang et al.’s finding that AR-FL knockdown activates the AKT/mTOR pathway, but in an AR-SV only expressing cell line, we found that AR knockdown led to the opposite regulation, AKT/mTOR pathway suppression [12]. Additionally, introducing an AR-SV in an AR-FL only expressing cell-line activated mTOR. This context specific AR-SV and AR-FL interaction with mTOR demonstrates the importance of considering AR-SVs in AR activity in HCC. Our data suggest AR-SVs in HCC do not simply function as surrogates for ligand bound AR-FL and warrant further study.

Interestingly, despite these similarities between HCC and PCa, there are several striking differences. AR-SV expression is thought to be upregulated in castration-resistant prostate cancer (CRPC) through selective pressure from androgen deprivation therapy (ADT) while HCC expresses AR-SVs in patients absent selective pressure from low levels of circulating androgens or anti-androgen therapy [20,44]. Additionally, enzalutamide, a competitive inhibitor of AR, shows clinical benefit in PCa patients initially until resistance develops, however, enzalutamide treatment showed no clinical benefit in HCC patients [26,45]. Finally, dysregulation of mTOR signaling is common in PCa, similar to HCC, with PTEN abnormalities being more common in PCa compared to other cancer types [46]. However, it has been reported that AR and mTOR interact differently in PCa than in HCC indicating a need for further research [12].

It is important to note that in the body of literature surrounding the role of AR in HCC, many immunoblotting techniques used to measure AR protein levels do not account for molecular weight (e.g., immuno-histochemistry) such that reagents targeting an epitope in the amino terminus of the AR are unable to distinguish AR-FL from lower molecular weight, carboxy-terminus truncated AR-SVs. Many of the most commonly used AR antibodies have amino terminus epitopes (Cell Signaling (#5153), EMD Millipore (AR-PG21), Abcam (ab74272, ab108341), Dako (AR441)) and have been used to assess AR levels within primary HCC tissue in multiple recent studies [6,12,21,32,36]. Given that many of these studies focus on previously unreported mechanisms of AR activation, it is important to keep in mind the potential contributions of unrecognized AR-SVs to composite AR activity in these studies.

### 4.4. Splicing Factor PRPF6 Associated with Increased AR Activity and Poor Prognosis

Song et al. provided additional insight into potential drivers of AR-SV expression in HCC. PRPF6, a precursor mRNA splicing factor, exhibits increased expression in HCC and its expression is correlated with both poor disease prognosis as well as HCC progression. PRPF6 increases AR transcription and by extension increases AR-SV expression through its interaction with both AR-FL and AR-SVs as most AR-SVs are thought to be alternatively spliced AR-FL transcripts. As such, PRPF6 plays a role in the upregulation of the transcription of several AR target genes. Further, this transcription enhancement via PRPF6 affects several genes related to the cell cycle ultimately leading to increased cellular proliferation within HCC [47]. This pathway provides insight into additional mechanisms of induced AR expression, but also how dysregulated splicing factors may increase AR activity and variant expression leading to increased AR oncogenic signaling.

## 5. AR-Targeted Therapeutic Strategies for HCC

### 5.1. Effectively Targeting the Androgen Receptor in HCC

Based upon the existence of constitutively active ligand-independent AR splice variants in HCC and multiple signaling pathways contributing to induced AR expression, effective therapeutic strategies to mitigate AR signaling in HCC will need to focus on targeting the AR itself as opposed to blocking androgen binding or suppressing androgen synthesis. Several different approaches have been considered including therapeutics binding the N-terminal domain (NTD) or DNA-binding domain (DBD) to either initiate protein degradation or to block AR transcriptional activity (Figure 1). Due to widespread AR-SV emergence within the context of CRPC [48], several new therapeutics are being developed to mitigate AR-SV specific or total AR signaling within the context of PCa that could be utilized for HCC.

### 5.2. Novel AR-SV-Targeted Agents

#### 5.2.1. DNA-Binding Domain-Targeted Agents

A very comprehensive overview of therapeutic approaches targeting AR was published recently by Xiang et al., which focused on small molecules that are thought to inhibit either AR-FL only or both AR-FL and AR-SVs [49]. A such, we focused mainly on agents designed to inhibit AR-SVs (Figure 5). Constitutively active AR functionality depends on the DBD and NTD [50,51,52]. The DBD is a well-resolved domain that binds with androgen response elements (AREs) and initiates AR-mediated transcription. It contains a P box (residues: 577–581) that interacts with the major groove of the DNA, and a zinc finger with the D box (residues: 596–600) that facilitates DBD-mediated AR dimerization. Lim et al. reported that Pyrvinium (a compound that usually exists in a pamoate salt form, also known as pyrvinium pamoate, or PP) inhibits AR activity by directly binding to the AR-DBD at the dimerization interface and the minor groove of ARE [53]. PP also showed in vitro inhibitory activities against AR-SVs, as well as in vivo potency against a 22Rv1 xenograft at 3 mg/kg dose [53]. Although the DBD is an attractive target for inhibiting both AR-FL and AR-SVs given its conserved nature and accessible structure, it’s challenging to obtain target specificity since the DBD among members of the nuclear receptor family including the progesterone receptor (NR3C3/PR), the glucocorticoid receptor (NR3C1/GR), and the estrogen receptor (NR3A1/ERα) is highly conserved [54,55,56]. To address this specificity issue, Li et al. and Dalal et al. identified a special residue (Gln592) in the AR-DBD active site which is specific to AR with a surrounding protein pocket that could be used to determine target selectivity [57,58]. They used virtual screening to search for compounds that are well-engaged with Gln592 and other surrounding residues within the pocket, and identified VPC-14228 and a synthetic analog, VPC-14449, that inhibit both AR-FL and AR-SVs through DBD binding [57]. Further publication from Dalal et al. reported VPC-17005 as an AR-DBD inhibitor that potentially disrupts dimerization of all AR isoforms [59]. Inspired by the promise of DBD inhibitors, Lee et al. reported a Proteolysis Targeting Chimeric (PROTAC) molecule, named MTX-23, using an AR-DBD binding motif to recruit von Hippel–Lindau (VHL) E3 Ubiquitin Ligase for targeted protein ubiquitination [60]. A similar approach was utilized by Bhumireddy et al. also using Compound 6 to degrade AR-SV protein by integrating AR-DBD and VHL ligands into a bifunctional PROTAC degrader [61]. Another PROTAC compound, ARD-61, has potent AR degrader activity mediated by LBD binding which has shown surprising benefit in CRPC even in the presence of AR-V7. The authors attribute this activity to the ability of AR-FL degradation to limit the activity of AR-V7, likely through limiting dimerization [62]. Additionally, the related compound ARV-110, a PROTAC showing AR degradation for both AR-FL and a variety of mutant AR types, has been in Phase 1/2 clinical trials showing efficacy against CRPC [63]. Taken together, developing small molecule inhibitors of protein dimerization or protein-nucleic acid interactions bring new insights to effectively combatting constitutively active AR and may be applicable to HCC as well.

#### 5.2.2. N-Terminal Domain-Targeted Agents

In addition to the DBD, the N-terminal domain (NTD) is another critical domain for constitutive activity and is located upstream of the DBD [64,65]. NTD is responsible for intra- and inter- molecular N/C interaction of AR and engagement with transcriptional machinery such as the RAP74 subunit of transcription factor II F which facilitates the initiation complex recruitment and binds to RNA polymerase II [66,67,68,69,70]. Blocking the function of the NTD is an attractive yet difficult approach since it is critical for both AR-FL and AR-SV transactivation, but its intrinsically disordered nature makes small molecule inhibitor development very challenging [65]. Sintokamide A (SINT1) is one of a few reported inhibitors of the AR-NTD. SINT1 is a natural product isolated from the marine sponge *Dysidea* sp., and it binds to the AR Activation Function-1 (AF-1) region of the NTD and inhibits the transactivation of both AR-FL and AR-SVs [71,72,73].

Another molecule from a marine sponge source called EPI-067 serves as the origin of a well-acknowledged family of AR-NTD inhibitors that covalently bind with the AF-1 region of AR-NTD [74,75]. EPI-001 was the pioneer molecule derived from EPI-067, and it binds to the transcription activation unit 5 (Tau-5) region of AF-1 through covalent interaction [76,77,78,79,80]. Derived from EPI-001, EPI-002 or “Ralaniten” is a stereospecific version [80,81,82,83]. The acetate prodrug of EPI-002 (EPI-506 or Ralaniten Acetate) is the first AR-NTD inhibitor to enter clinical trials (NCT02606123) [84,85,86,87]. The phase I trial of Ralaniten Acetate showed evidence of PSA reduction in some patients receiving higher doses, but the trial was halted due to excessively high pill burden [65]. Further prodrug development resulted in EPI-7386 providing improved metabolic stability and pharmacokinetic properties. The clinical trial of EPI-7386 (NCT04421222) was initiated in 2020, and is currently recruiting patients for a Phase I study [64,88,89]. Inspired by the EPI series, Ban et al. designed the NTD-targeting compound VPC-220010 [90]. VPC-220010 not only inhibits the transcriptional activities of both AR-FL and AR-SVs, but also decreases the DNA binding of the AR to AR-regulated genes resulting in reduced AR-mediated transcription. Taken together, the EPI series established proof-of-concept AR-NTD inhibition and demonstrated that targeting of constitutively active AR-SVs was possible.

Another group of compounds that interfere with AR-SVs are the UT-series, also known as selective androgen receptor degraders (SARDs). Hwang et al. designed a compound named UT-69 by combing the structural features of an AR antagonist and an AR agonist (Enobosarm), which showed efficacy against Enzalutamide-resistant xenografts and degraded both AR-SVs and AR-FL [91]. Further development of UT-69 resulted in indolyl and indolinyl classes of analogs, from which UT-155 is the representative lead compound out of extensive structure–activity relationship (SAR) exploration [91]. Interestingly, UT-69 and UT-155 not only degrade AR but also show binding affinity with AR-LBD. Protein NMR study also confirmed that UT-155 also binds to the AF-1 region between residues 244 and 360 at AR-NTD. Interestingly, the enantiomer (R)-UT-155 only interacts with AF-1, but not LBD [91,92], and after replacing the indolyl moiety of UT-155 with halogen-substituted pyrazole or triazole, new molecules on a novel scaffold named UT-34 [93], compound 26a [94], and compound 26f [95] also displayed potent AR-FL and AR-SV degradation activities.

Collectively, there have been a number of therapeutic approaches targeting ligand-independent AR activity. AR-SVs that lack the ligand-binding domain can be deactivated through DBD blockade, NTD inhibition or ubiquitin–proteasome pathway-mediated protein degradation. Given the clinical relevance of AR-SVs to HCC, it is reasonable to believe that these therapeutic approaches would also be applicable to HCC patients.

### 5.3. Repurposed

Beyond the novel compounds currently being developed, there have been several proposals to repurpose known compounds. One study found that quercetin [96], a naturally occurring poly-phenol found in fruits and vegetables, reduced AR-SV and AR-FL expression through mitigating AR synthesis and, when used in combination with enzalutamide, was able to improve sensitivity in enzalutamide resistant prostate cancer cells [48].

Another leading candidate for repurposing is the anti-helminthic drug, niclosamide [97]. Lui et al. performed a drug screen looking for compounds capable of inhibiting AR-V7 activity encompassing 1120 FDA approved drugs. They found niclosamide to selectively inhibit AR-V7 and found that niclosamide was able to inhibit both PCa cell proliferation and in vivo tumor growth [98]. From a mechanistic standpoint, niclosamide is thought to indirectly induce AR-SV degradation through the ubiquitin–proteolysis pathway rather than through direct interaction with the AR [98,99]. Additionally, niclosamide’s potency and anti-cancer effects have also been linked to a pH dependent mechanism relating to niclosamide’s properties as a protonophore, denoting compounds able to transport protons across a cellular membrane [100]. A separate drug screen carried out by Chen et al. assembled a representative profile of mRNA expression in HCC and then looked for existing drugs with opposing effects on gene transcription in HCC cells. From this pseudo-phenotypic approach, niclosamide emerged as a top hit. They thoroughly investigated niclosamide along with its ethanolamine salt (NEN) as an anti-HCC drug in both genetic and patient derived xenograft models of HCC. NEN was effective in slowing tumor growth in vivo and was able to reverse gene signatures of key signaling pathways such as AKT-mTOR and EGFR-Ras-Raf [101]. The convergence of these two screening approaches supports the potential of niclosamide in HCC as it is both effective against AR-SVs and generally has anti-HCC activity.

Based upon its promising anti-PCa effects, niclosamide has been evaluated in PCa clinical trials. A phase I trial combining niclosamide with enzalutamide in CRPC was halted due to the inability to achieve therapeutically efficacious niclosamide levels in plasma at the maximum tolerated dose [102]. This highlights a key problem with utilizing niclosamide as an anti-cancer drug: poor systemic absorption. Originally designed to treat gut parasites where absorption was undesirable, niclosamide is well known to have poor drug-like properties which limit its use as a systemic therapy [103]. A later Phase Ib trial found that abiraterone/prednisone in combination with reformulated version of niclosamide, designed to improve its systemic absorption, was able to achieve therapeutic levels of niclosamide without encountering dose limiting toxicities. Additionally, they saw a PSA response in five of their eight enrolled patients, suggesting clinical anti-PCa efficacy [104].

Additional efforts have been made to improve niclosamide for use in PCa including the exploration of several analogs with the goal of both improving niclosamide’s bioavailability and its anti-AR potency. One analog, ARVib-7 (Figure 6), showed a 6-fold improvement over niclosamide in both maximal plasma concentration (Cmax) and total systemic exposure in rats following an oral dose. Importantly, ARVib-7 was similar to or improved compared to niclosamide’s efficacy in degrading AR-V7 and inhibiting PCa growth in vitro and in vivo [99]. These reported improvements for niclosamide and its analogs in PCa support a similar approach in AR positive HCC.

### 5.4. Combined mTOR and AR Inhibition

In light of the key role that the mTOR pathways plays in HCC and its interaction with AR signaling, Zheng et al. provide strong support for a therapeutic strategy combining both AR and mTOR inhibition [2]. They showed that AR knockdown resulted in mTOR activation, so utilizing a combined inhibition strategy may help to mitigate the pro-cancerous effects of both pathways. However, given the evidence showing that AR-SVs may not interact with mTOR in the same way as AR-FL, further exploration of AR and mTOR cross talk is warranted and is key to developing an effective combination strategy targeting both signaling pathways. A strategy accounting for the need to inhibit or degrade both AR-FL and AR-SVs as well as carefully exploring the resulting impacts upon mTOR in both AR-SV positive and AR-SV negative disease may prove beneficial in treating HCC.

## 6. Conclusions

To conclude, AR signaling plays a key role in HCC progression and the sexual dimorphism of the disease; however, therapeutically targeting the AR-axis has thus far been unable to effectively treat HCC due to the combined effects of induced AR expression, post-translational ligand-independent AR-activation, and the presence of ligand-independent constitutively active AR-SVs. As HCC is a widespread and genetically heterogeneous disease lacking targeted therapy, AR provides a key therapeutic opportunity. The presence of abundant AR-SVs provides an additional explanation for previous failures in anti-androgen or hormone ablative approaches and may also explain, in part, difficulty in targeting key signaling pathways within HCC such as AKT-mTOR. Developing therapeutics that target both AR and AR-SVs could provide an improved therapeutic option for HCC patients with AR positive disease.

While considerable work has been carried out to understand the role of AR within HCC, additional work is needed to better understand the influence of AR-SVs on known oncogenic signaling in HCC, their potential for HCC-specific action as compared to PCa, and potentially distinct AR-SV and AR-FL genetic programs in HCC. Unlike PCa, AR-SVs expressed in HCCs are not an adaptive response to therapeutic pressure from anti-androgen treatment as HCC patients are not routinely administered anti-androgens. AR-SVs in HCC are likely distinct in origin such that AR-SV expression profiles differ between PCa and HCC and may even contain novel HCC-specific variants. Better defining the “AR spliceosome” in HCC would facilitate development of biomarkers for AR activity and could help improve patient selection for future AR-targeted therapies. Despite promising recent advances in HCC treatment with immunotherapies [105], sexual dimorphism remains a resolute feature of HCC [106], supporting the continued development of effective AR-targeted approaches for HCC.

## Figures and Tables

**Figure 1 ijms-23-13768-f001:**
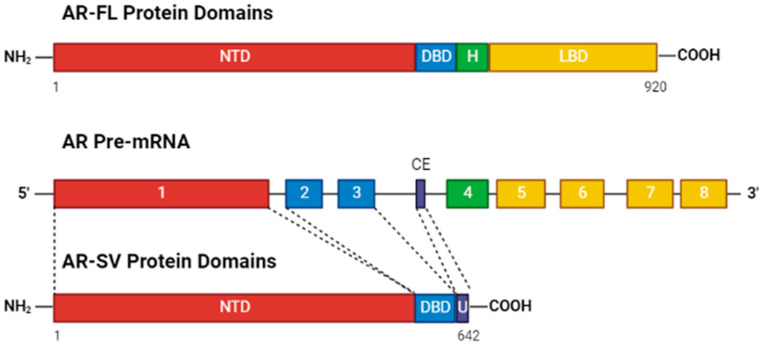
Diagram of AR protein domains and pre-mRNA and its eight exons. AR-FL contains an N-terminal domain (NTD), DNA-binding domain (DBD), a Hinge region, and a ligand-binding domain (LBD) which binds androgen to facilitate nuclear localization, dimerization, and binding via the DBD to AREs within the DNA. However, in alternative splicing, a cryptic exon (CE) between exons 3 and 4 is incorporated leaving AR-SVs without the hinge region and the LBD, making a constitutively active ligand-independent variant of the AR. Figure created in BioRender.

**Figure 2 ijms-23-13768-f002:**
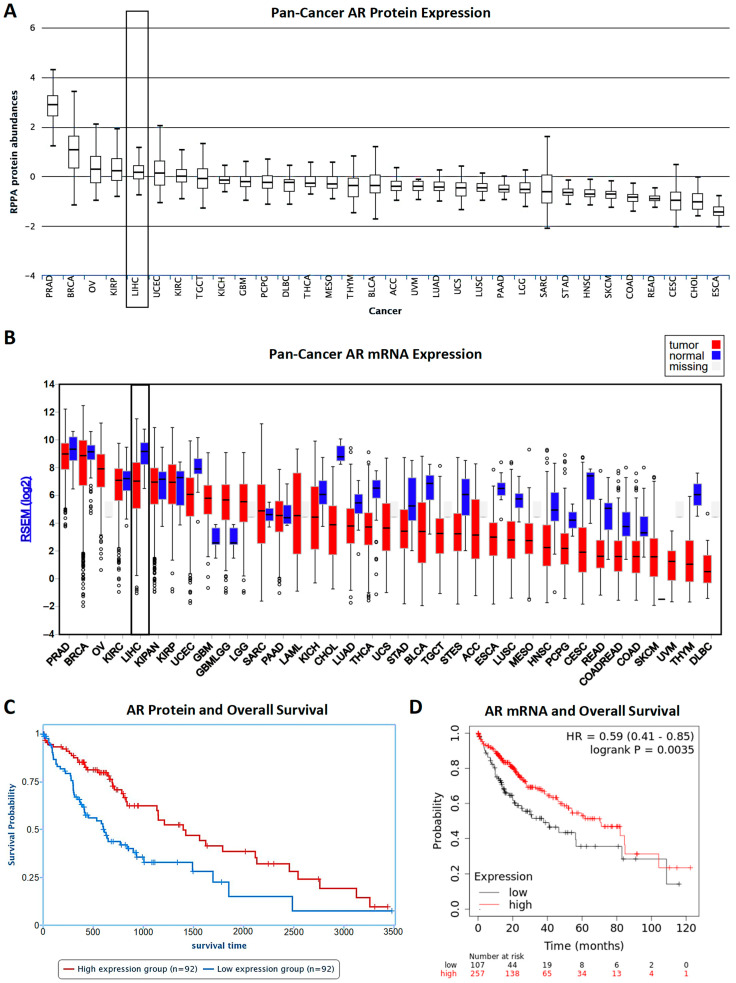
Plots showing AR mRNA and protein expression in HCC. (**A**) Figure charting AR protein expression from highest to lowest across cancer types from the TCGA as measured by reverse-phase protein assay on TCGA tissue samples using AR antibody (ab52615 by Abcam). Data encompass 184 liver cancer patients. LIHC (Liver Hepatocellular Carcinoma) data are highlighted. Figure generated in The Cancer Protein Atlas (TCPA) (https://www.tcpaportal.org/) [15,16]. (**B**) Figure charting AR mRNA expression from highest to lowest across TCGA cancer types as determined using RNA-Seq by Expectation-Maximization (RSEM) from TCGA data. LIHC (Liver Hepatocellular Carcinoma) data are highlighted and encompass 363 liver cancer patients and 50 normal samples. Figure generated by FireBrowse (http://firebrowse.org/) from the Broad Institute. (**C**) Kaplan–Meier plot showing the impact of AR protein expression on overall survival in HCC. Higher AR expression is correlated with better overall survival. Data encompass a total of 184 liver cancer patients and survival time is notated in days. Log-rank *p* value is 0.00063428. Figure generated in The Cancer Protein Atlas (TCPA) [15,16]. (**D**) Kaplan–Meier plot showing impact of AR mRNA expression on overall survival. Higher AR expression is correlated with better overall survival when compared to patients with lower AR expression. Data encompass a total of 364 liver cancer patients. Figure created in Kaplan-Meier Plotter (https://kmplot.com/) [14,17].

**Figure 3 ijms-23-13768-f003:**
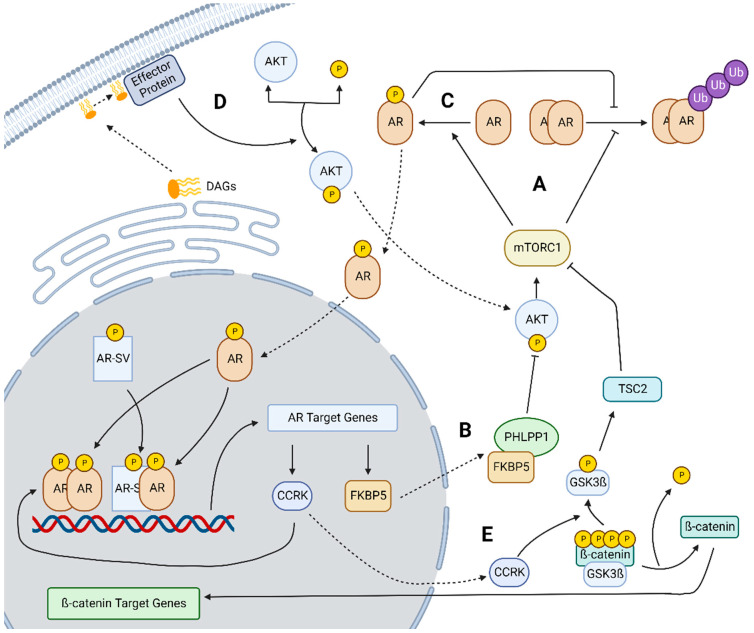
Map of mechanisms of inducing AR expression in HCC showing linkages to mTOR, lipogenesis, and CCRK-mediated signaling. The splice variants shown demonstrate a non-androgen-mediated pathway of AR signaling, however, the effects of AR-SV signaling may not be directly in line with AR-FL signaling and further study is warranted to elucidate the impacts of AR-SVs on these pathways. (A) mTORC1 interacts with AR by both inhibiting AR degradation and upregulating AR activity. (B) AR-FL target gene FKBP5 scaffolds onto PHLPP1, an AKT phosphatase, to inhibit AKT phosphorylation by extension mTORC1 activity forming a feedback loop. (C) mTORC1 upregulates AR activity by phosphorylating AR at serine 96. (D) Specific DAGs from lipogenesis can increase Akt activity through binding at an unknown effector protein leading to increased AKT -mediated AR activity. (E) CCRK phosphorylates GSK3ß deactivating it, which leads to both activation of β-catenin and inhibition of TSC2 by preventing its phosphorylation by GSK3ß. Figure created in BioRender.

**Figure 4 ijms-23-13768-f004:**
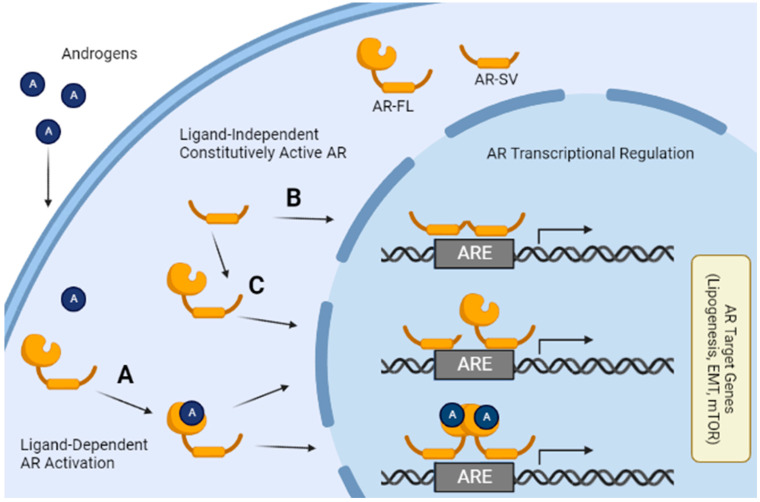
Illustration of mechanisms of ligand-independent and ligand-dependent AR-mediated transcriptional regulation. (A) Androgens bind to the LBD of AR-FL leading to nuclear localization, dimerization, and binding of the DBD to androgen response elements (AREs) in the DNA. (B) In the absence of androgens AR-SVs both dimerize and localize to the nucleus. (C) In the absence of androgens, AR-SVs are also able to facilitate AR-FL nuclear localization without androgen binding to the LBD. These various methods of AR activation ultimately result in the regulation of AR target genes and more broadly gene pathways such as lipogenesis, EMT, and mTOR-related pathways. Figure created in BioRender.

**Figure 5 ijms-23-13768-f005:**
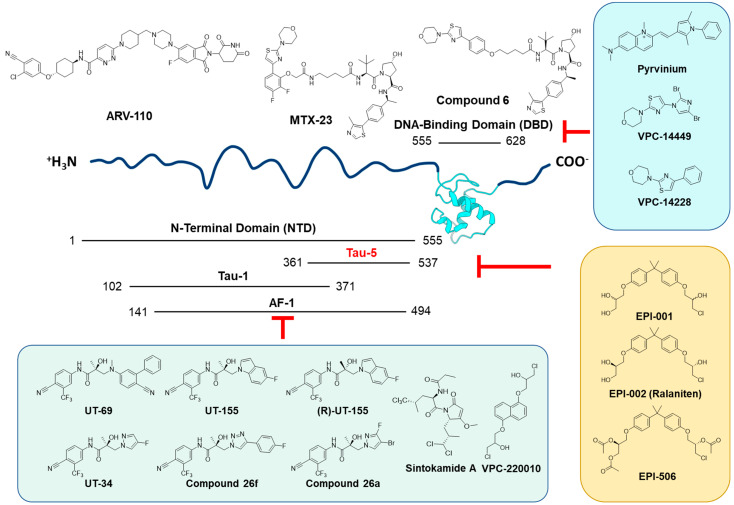
Schematic illustration of androgen receptor splice variant from N-terminal domain to C-terminal. Compounds that inhibit the AF-1 region of NTD domain are listed in the lower left hand green highlighted frame, compounds that inhibit the Tau-5 region of the NTD are listed in the yellow highlighted frame, and compounds inhibiting the DBD are listed in the upper right blue highlighted frame. Two reported PROTAC molecules based on DBD inhibitors are shown on the top of the figure.

**Figure 6 ijms-23-13768-f006:**
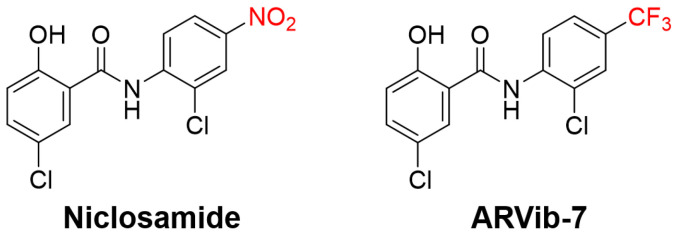
Structure of niclosamide and its analog, ARVib-7.

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
