# Peer review of "Constitutively Active Androgen Receptor in Hepatocellular Carcinoma"

_ijms, 2022, doi:10.3390/ijms232213768_

Round 1

Reviewer 1 Report

No comments. The review is well written and should be published. I have just one minor inquiry. Is there any possibility authors could provide a schematic about Androgen dependent and Androgen independent signaling in HCC?

Reviewer 2 Report

Montgomery et al. present a rather comprehensive and insightful review on the role of Androgen Receptor in HCC and potential mechanisms of action in tumorigenesis and resulting etiology. The authors note upfront the intriguing point that HCC has a higher prevalence in males. The review is well written, very informative and certainly of interest to the community given the rather poor prognoses associated with HCC.

-          The authors are encouraged to include data from TCGA/COSMIC showing AR gene amplification across different cancer types and mRNA expression in tumor biospies vs normal biopsies. Kaplan-Meier data showing AR-dependent prognoses would be valuable to include and draw the point to prognoses in high vs low AR cases. A number of web-based tools are available to freely explore and download such data in figure-ready formats. E.g. KM plotter etc.

-          Please consider including a domain structure figure of the AR receptor, showing the different domains and noting the respective biochemical function. This would make it easier to refer certain sections of the text that speak on N-terminal binding domain etc.

-          Section 1.1: It would be ideal to include additional details on HCC statistics such as age of HCC onset, 5-year survival rates for men and women post diagnosis, current and projected incidence and mortality rates. Refer to SEER, CDC  

-          Section 1.3, Line 64: There is also other direct evidence that AR modulates EMT gene expression, although in differed cell types. E.g. PMID 18794357 and 25307492. Please include this in the section.

-          Section 1.4: Inhibition of androgen biosynthesis showed no therapeutic benefit. But has direct inhibition of AR itself shown to result in tumor regression either in clinical trials or preclinical models? E.g. PMID 31931431. It would be good to make a brief note even if known in a different cancer type. Perhaps ideal to also include relevant details on AR degraders (ARV-110) in section 5.2.1.

-          Line 114: “Androgen-independent AR overexpression”: Does androgen stimulate both transcriptional activity and expression of AR? Please clarify /rectify.

-          Are there known interactors of the AR protein, as determined by proteomic approaches? A brief paragraph on this and an accompanying table listing all known interactors in cell/tissue type along with references would add value to the section 2.

-          One wonders if discrepant AR expression in between males and females is a primary cause of higher HCC incidence in males. Can the authors perhaps explore/mine AR expression data from the Sex Associated gene Database (SAGD) to shed light on this?
